# Meteorological controls on atmospheric particulate pollution during hazard reduction burns

Giovanni Di Virgilio[1], Melissa Anne Hart[1,2], Ningbo Jiang[3]

[1]Climate Change Research Centre, University of New South Wales, Sydney, 2052, Australia
[2]Australian Research Council Centre of Excellence for Climate System Science, University of New South Wales, Sydney, 2052, Australia
[3]New South Wales Office of Environment and Heritage, Sydney, 2000, Australia

*Correspondence to*: Giovanni Di Virgilio (giovanni@unsw.edu.au)

**Abstract**. Internationally, severe wildfires are an escalating problem likely to worsen given
projected changes to climate. Hazard reduction burns (HRB) are used to suppress wildfire
occurrences, but they generate considerable emissions of atmospheric fine particulate
matter, which depending upon prevailing atmospheric conditions, can degrade air quality.
Our objectives are to improve understanding of the relationships between meteorological
conditions and air quality during HRBs in Sydney, Australia. We identify the primary
meteorological covariates linked to high $PM_{2.5}$ pollution (particulates < 2.5 μm diameter)
and quantify differences in their behaviours between HRB days when $PM_{2.5}$ remained low,
versus HRB days when $PM_{2.5}$ was high. Generalised additive mixed models were applied to
continuous meteorological and $PM_{2.5}$ observations for 2011-2016 at four sites across
Sydney. The results show that planetary boundary layer height (PBLH) and total cloud cover
were the most consistent predictors of elevated $PM_{2.5}$ during HRBs. During HRB days with
low pollution, the PBLH between 00:00 and 07:00 h (local time) was 100-200 m higher than
days with high pollution. The PBLH was similar during 10:00-17:00 h for both low and high
pollution days, but higher after 18:00 h for HRB days with low pollution. Cloud cover,
temperature and wind speed reflected the above pattern, e.g. mean temperatures and wind
speeds were 2 °C cooler and 0.5 m s$^{-1}$ lower during mornings and evenings of HRB days
when air quality was poor. These cooler, more stable morning and evening conditions
coincide with nocturnal westerly cold air drainage flows in Sydney, which is associated with
reduced mixing height and vertical dispersion, leading to the build-up of $PM_{2.5}$. These
findings indicate that air pollution impacts may be reduced by altering the timing of HRBs by
conducting them later in the morning (by a matter of hours). Our findings support location-
specific forecasts of the air quality impacts of HRBs in Sydney and similar regions elsewhere.

## 1 Introduction

Many regions experience regular wildfires with the potential to damage property, human health, and natural resources (Attiwill and Adams, 2013). Internationally, the frequency and duration of wildfires are predicted to increase by the end of the century (e.g. Westerling et al., 2006; Flannigan et al., 2013). Wildfire frequency and duration have increased in western North America since the 1980s (Westerling, 2016). Their frequencies have also increased in south-eastern Australia over the last decade (Dutta et al., 2016), with a predicted 5-25 % increase in fire risk by 2050 relative to 1974-2003 (Hennessy et al., 2005), a risk compounded by climate change (Luo et al., 2013). In an effort to mitigate the escalating wildfire risk, fire agencies in Australia, as is the case internationally, conduct planned hazard reduction burns (HRBs; also known as prescribed or controlled burns). HRBs reduce the vegetative fuel load in a controlled manner and aim to lower the severity or occurrence of wildfires (Fernandes and Botelho, 2003).

Both wildfires and HRBs generate significant amounts of atmospheric emissions such as particulate matter (PM), which can impact urban air quality (Keywood et al., 2013; Naeher et al., 2007; Weise et al., 2015), and consequently public health (Morgan et al., 2010; Johnston et al., 2011). Of particular concern are fine particulates with a diameter of 2.5 μm or less, '$PM_{2.5}$'. Increased $PM_{2.5}$ concentrations are related to health effects including lung cancer (Raaschou-Nielsen et al., 2013) and cardiopulmonary mortality (Cohen et al., 2005). These impacts can be more severe for vulnerable groups, like the young (Jalaludin et al., 2008), elderly (Jalaludin et al., 2006) and individuals with respiratory conditions (Haikerwal et al., 2016).

Sydney, located in the south-eastern Australian state of New South Wales (NSW), is the focus of this study because HRBs make a significant contribution to PM pollution in this city and the surrounding metropolitan region (Office of Environment and Heritage, 2016). Sydney is Australia's largest city with 4.9M inhabitants (ABS, 2016). Approximately 130,911 ha in NSW was treated by HRBs during 2014-15 (RFS, 2015) and this figure is projected to increase annually (NSW Government, 2016). Smoke events between 1996 and 2007 in Sydney attributed to wildfires or HRBs were associated with an increase in emergency department attendances for respiratory conditions (Johnston et al., 2014). Hence, a potential consequence of HRBs is that Sydney's population experiences poor air quality and

its associated health impacts (Broome et al., 2016). Furthermore, the eastern Australian fire season is projected to start earlier by 2030 under future climate change (Office of Environment and Heritage, 2014). This could restrict the period within which HRBs can occur, potentially exposing populations to particulates over more concentrated time-frames.

Sydney is located in a subtropical, coastal basin bordered by the Pacific Ocean to the east and the Blue Mountains 50 km to the north-west (elevation 1189 m, Australian Height Datum). Its air quality is influenced by mesoscale circulations, such as terrain-related westerly drainage flows in the evening, and early morning, easterly sea breezes in the afternoon (Hyde et al., 1980). These processes interact with synoptic-scale high-pressure systems (Hart et al., 2006). A recent study by Jiang et al. (2016b) further examined how synoptic circulations influence mesoscale meteorology and subsequently air quality in Sydney. The results showed that smoke generated by wildfires and HRBs makes a significant contribution to elevated PM levels in Sydney, in particular, under a combined effect of typical synoptic and mesoscale conditions conducive to high air pollution. However, analysis of the local (i.e. city-scale) meteorological processes that influence air quality during HRBs is still sparse. Previous research focusing on a single site in Sydney found that $PM_{2.5}$ concentrations were higher during stable atmospheric conditions and on-shore (easterly) winds (Price et al., 2012). Elsewhere, $PM_{2.5}$ concentration was mainly influenced by the receptor-to-burn distance and wind hits during HRBs (Pearce et al., 2012). We therefore have three aims: 1. summarise the temporal variation in $PM_{2.5}$ concentrations in Sydney and how this relates to HRB occurrences; 2. characterise $PM_{2.5}$ pollution sensitivities to meteorological and HRB variables to identify the primary covariates connected to high pollution; 3. identify the differences in covariate behaviours between HRB days when $PM_{2.5}$ pollution is low, versus burn days when pollution is high. Achieving these aims will help efforts to forecast the air pollution impacts of HRBs in Sydney, and more broadly, in Australia or elsewhere in the world.

## 2 Data

### 2.2 Meteorological, air quality and temporal variables

Continuous time series of hourly meteorology and $PM_{2.5}$ (µg m$^{-3}$) observations between January 2005 and August 2016 inclusive were obtained from four air quality monitoring stations (Chullora, Earlwood, Liverpool and Richmond) in the NSW Office of Environment and Heritage (OEH) network in Sydney (Fig. 1). Monitoring stations are located at varying elevations and in semi-rural, residential and commercial areas (Table 1). These four locations were chosen because they have the longest, uninterrupted record of $PM_{2.5}$ measurements in Sydney. Prior to 2012 $PM_{2.5}$ was measured using tapered element oscillating microbalance (TEOM) systems. Since 2012 beta attenuation monitors (BAM) have been used to measure $PM_{2.5}$. Although there appear to be effects from instrument change, such effects are generally small if compared to the daily-to-day or hourly fluctuations in $PM_{2.5}$ levels.

To compare how $PM_{2.5}$ concentrations varied over daily and monthly timescales, we also obtained hourly measurements of $PM_{10}$ (µg m$^{-3}$), nitrogen dioxide ($NO_2$) (parts per hundred million - pphm) and oxides of nitrogen ($NO_X$) (pphm) from these stations. Meteorological variables included in our analyses were: surface wind speed (m s$^{-1}$), wind direction (°), surface air temperature (°C) and relative humidity (%). Hourly global solar radiation (W m$^{-2}$) data were available at the Chullora station only, but were subsequently omitted as a predictive variable (see: 3.3.1 Model selection).

Hourly total cloud cover (okta) and mean sea level pressure (MSLP; hPa) were obtained from the Australian Bureau of Meteorology (BoM) Sydney Airport weather station (WMO station number 94767). These are included as covariates in models for the four monitoring sites. Twenty-four hour rainfall totals (mm) were approximated for each OEH station from the BoM weather station that is nearest (Fig. 1).

Given its role in the turbulent transport of air pollutants (Seidel et al., 2010; Pal et al., 2014; Sun et al., 2015; Miao et al., 2015), we included planetary boundary layer height (PBLH) as an explanatory variable. PBLH has previously been derived from observational meteorological data by Du et al. (2013) and Lai (2015), using a method which they found was an effective estimate of the PBLH and its relationship with PM concentrations. Although direct PBLH measurements would be ideal, these are unavailable for the study domain at

appropriate spatial and temporal resolutions. Hence, we derived PBLH estimates at the
location of each monitoring station from a subset of the meteorological data following the
method used by the above authors (Eq. (1) and Eq. (2)).

$$PBLH = \frac{121}{6}(6-s)(t-td) + \frac{0.169s(ws+0.257)}{12f\ln(\frac{h}{l})} \tag{1}$$


$$f = 2\Omega\sin\theta \tag{2}$$


where $s$ is a stability class that estimates lateral and vertical dispersion; $t$ is surface air
temperature and $td$ is surface dew point temperature (approximated for the location of
each station using the method proposed by Lawrence (2005)); $ws$ is wind speed; $h$ is wind
speed altitude in m for a given monitoring station; $l$ is the station's estimated surface
roughness index, $f$ is the Coriolis parameter in s$^{-1}$; $\Omega$ is the earth's rotational speed (rad s$^{-1}$)
and $\theta$ is the station latitude. The stability typing scheme was based on the Pasquill-Gifford
(P-G) stability categories (Turner, 1964), via a turbulence-based method using the standard
deviation of the azimuth angle of the wind vector and scalar wind speed.
We calculated the 24-hour mean for hourly meteorological and PM$_{2.5}$
measurements, where wind direction was vector-averaged (i.e. averaging the u and v wind
components). Log-transformations were applied to PM$_{2.5}$ and rainfall. Applying
transformations to the remaining explanatory variables did not greatly reduce
heterogeneity.
Temporal variables trialled for inclusion in analyses included day of the year,
weekday, week, month (all representing different seasonal terms) and year (because air
quality varies from year to year). A Julian date variable was incorporated to represent the
longer-term trend in PM$_{2.5}$ concentrations.
**2.3 Burns**
Historical records of HRBs conducted between January 2005 and August 2016 in NSW were
obtained from the NSW Rural Fire Service (RFS), the firefighting agency responsible for the
general administration of HRBs. There were a total of 9200 fire polygons in this data set
prior to data conditioning (see: 3 Methods). HRBs are conducted predominantly in Autumn
(months of March to May in the southern hemisphere) and Spring (September to

November), and often at weekends, typically, with burns lit in the early morning. Most historical HRBs have occurred to the west and north-west of Sydney (Fig. 2). Additional predictive variables derived from the HRB data (all daily values) were: total number of burns, total burn surface area (ha), median burn elevation (m), median fire duration (days) and median fire distance from the geographic centre of the monitoring stations (km).

It is important to note that other potential sources of $PM_{2.5}$ emissions in Sydney include motor vehicles, soil erosion and occasional dust storms. Use of domestic wood-fired heaters can also make a substantial contribution to $PM_{2.5}$ concentrations during Winter months (June to August), which is when HRBs are generally not conducted. However, between 2011 and 2016, average $PM_{2.5}$ air quality index (AQI) values were higher on days when either HRBs or wildfires occurred relative to days when there were no fires (Fig. 3).

## 3 Methods

### 3.1 Statistical approach: generalised additive mixed models

Generalized additive models (GAMs) (Hastie and Tibshirani, 1990) offer an appropriate approach with respect to air quality research because relationships between covariates are often non-linear, an issue which can be addressed within the GAM framework. In addition to the seasonal pattern of hazard reduction burning, $PM_{2.5}$ concentrations in Sydney also show daily, monthly, seasonal and annual variation. Adding terms to a GAM to account for these temporal variations fails to deal with residual autocorrelation completely, as is evident in the autocorrelation function (ACF) of the residuals (Fig. S1, Supplementary Material). Given the residual autocorrelation and non-independence of the data, we used a generalised additive mixed modelling (GAMM) approach to take account of the seasonal variation and trends in the data. GAMMs can combine fixed and random effects and enable temporal autocorrelation to be modelled explicitly (Wood, 2006). We assumed a Gaussian distribution and used a log link function. Cubic regression splines were used for all predictors except wind direction and day of year which used cyclic cubic regression splines, because there should be no discontinuity between values at their end points. Experimenting with alternative smooth classes did not drastically affect model results or diagnostics. Smoothing parameters were chosen via restricted maximum likelihood (REML). We implemented GAMMs with a temporal residual auto-correlation structure of order 1 (AR-1).

More complex structures (e.g. auto-regressive moving average models; ARMA) of varying
order or moving average parameters produced marginally higher Akaike information criteria
(AICs) (e.g. mean = 259.6) than models with AR-1 auto-correlation (mean AIC = 259.02).
Omitting a correlation structure entirely produced the largest AICs (mean AIC = 279.5). In all
cases, the AR models for the residuals were nested within month (nesting within week and
year was also trialled, but produced higher AICs). Auto-correlation plots obtained by
applying the GAMMs using the AR-1 structure showed that short-term residual
autocorrelation in the residuals had been removed relative to using GAMs (Fig. S1-2 in
Supplementary Material).
**3.2 PM$_{2.5}$ trend estimates, monthly and daily means**
We first used the GAMM framework to estimate the annual trend in the weekly mean
concentrations of PM$_{2.5}$ for 2005–2015, split by season, with Julian day as the only predictor.
Monthly and daily mean PM$_{2.5}$, PM$_{10}$, NO$_2$ and NO$_X$ concentrations for all years were also
compared to assess how concentrations of each pollutant varied with these timescales.  The
latter analyses were performed using *R* software for statistical computing (R Development
Core Team, 2015) and the *openair* package (Carslaw and Ropkins, 2012). The annual trend
and subsequent statistical analyses described below were performed using *R* software and
packages *mgcv* (Wood, 2011) and *nlme* (Pinheiro et al., 2017).
**3.3 Identifying the meteorological and burn variables related to elevated PM$_{2.5}$**
To assess how PM$_{2.5}$ concentrations vary in relation to the meteorological, burn and
temporal variables, the GAMMs were applied to each monitoring site separately and
focused on the period January 2011-August 2016. There were comparatively fewer HRBs
conducted prior to 2011, hence the choice of this timeframe. For each station, we split the
data into two subsets: 1) for all days when HRBs were conducted and the PM$_{2.5}$
concentration was less than the median PM$_{2.5}$ concentration for the location in question,
'low pollution days'; 2) for all HRB days when the PM$_{2.5}$ concentration was greater than the
median value for the location in question, 'high pollution days' (the minimum/maximum
number of observations in each low/high subset was in the range 179-189). The time series
were conditioned in this manner to better characterise the differences in covariate
behaviours between burn days when pollution remains low versus burn days and elevated
PM$_{2.5}$. Since our focus is specifically on PM$_{2.5}$ concentrations during HRBs, days when
wildfires had occurred were excluded.

*3.3.1 Model selection*

Using the GAMM framework described above, we started with a model where the fixed
component included all predictive variables. We used variance inflation factors (VIF) to test
variables for collinearity (Zuur et al., 2010). We sequentially dropped covariates with the
highest VIF and recalculated the VIFs, repeating this process until all VIFs were smaller than
a threshold of 3.5. This VIF threshold was selected as a compromise between the thresholds
of 3 and 10 stipulated in Zuur et al. (2010). Following this process, explanatory variables
were dropped from the initial model if they were not statistically significant in any case. As a
result, global solar radiation, relative humidity, burn elevation, burn duration, day of year,
weekday, week and year were excluded.
An intermediate model included HRB distance as a covariate. Exploratory GAMM
analyses using this model configuration revealed that on average, beyond a distance of ca.
300 km, the influence of prescribed burns on PM$_{2.5}$ concentrations at the target locations
was negligible (Fig. S3, Supplementary Material). Subsequent models excluded burn
distance and burns > 300 km from the geographic mean centre of the monitoring stations.
Hence, the fixed component of our optimal model used the following predictors: PBLH,
MSLP, temperature, total cloud cover, rainfall, wind speed, wind direction, number of burns
per day, total area burnt per day and Julian day.

**3.4 Diurnal variation in relation to elevated PM$_{2.5}$**

Meteorological covariates relevant to high PM$_{2.5}$ concentrations were identified via the
GAMMs based on criteria of statistical significance at more than one location, or where the
influence of covariates on PM$_{2.5}$ showed a marked distinction between pollution conditions.
We then used the hourly meteorological data for these select covariates to compare their
mean diurnal variation on burn days with low versus high pollution. The 95 % confidence
intervals of these diurnal means were calculated using bootstrap re-sampling with 1000
replicates.

## 4 Results

### 4.1 Temporal variation in PM$_{2.5}$ concentrations

There is an increasing inter-annual trend in weekly mean PM$_{2.5}$ concentrations in all seasons during 2011 to 2015, especially in summer and winter (Fig. 4). Mean PM$_{2.5}$ concentrations range from 6 - 10 µg m$^{-3}$. Mean monthly PM$_{2.5}$ averaged over all years shows increasing concentrations from early autumn (March), peaking in May, then decreasing towards the end of winter, before increasing again from early spring (Fig. 5a). Notably, mean daily PM$_{2.5}$ concentrations (averaged over all years) are higher at weekends relative to other pollutants (PM$_{10}$, NO$_2$ and NO$_x$; Fig. 5b).

### 4.2 Meteorological and burn variables related to PM$_{2.5}$

Adjusted $R^2$ values for high pollution models were between 0.44 and 0.60, and between 0.29 and 0.39 for the low pollution models (Table 2). PBLH and total cloud cover were the most consistent predictors of elevated PM$_{2.5}$ during HRBs (Table 2). On high pollution days, PBLH had a statistically significant, negative influence on predicted PM$_{2.5}$ concentrations at all locations (Fig. 6). This influence was generally more linear on high pollution days, relative to low pollution days. Notably, fitted curves for PM$_{2.5}$ – PBLH were steeper at lower altitudes (< 800 m) in the high pollution condition. Cloud cover had a negative influence on predicted PM$_{2.5}$ concentrations that was significant in all but one case (Table 2), though fitted curves do not appear to differ noticeably between pollution conditions (Fig. 7). Although temperature and wind speed showed a more variable pattern of statistical significance (Table 2), they exhibited marked differences in behaviour between low and high pollution days. During high pollution, temperature typically had a negative, curvilinear influence on fitted PM$_{2.5}$ values (Fig. 8). This negative influence flattens or reverses at temperatures > 20 °C. In contrast, the PM$_{2.5}$ – temperature relationship was weak and linear during low pollution days. Wind speed had a significant influence on PM$_{2.5}$ only at Earlwood and Richmond (Table 2). During low pollution days, this association is negative at most locations. During high pollution conditions at Chullora and Earwood, there is a positive influence on PM$_{2.5}$ at low wind speeds which reverses at speeds above ca. 2 m s$^{-1}$ (Fig. S7, Supplementary Material). During HRBs and high pollution, wind direction curves show peaks between approximately 250 and 310 degrees at Chullora, Earlwood and Liverpool (south-westerly to

north-westerly flows) (Fig. 9). Earlwood frequently experiences north-westerly flows during
Spring, Autumn and Winter, whilst south-westerly flows are common during the same
seasons at Liverpool (Fig. S4, Supplementary Material).
The remaining meteorological predictors either did not show marked differences
between pollution conditions or were statistically significant in only one instance. Rainfall
generally had a negative influence on $PM_{2.5}$ during HRBs (Fig. S5, Supplementary Material).
MSLP had a positive association with higher $PM_{2.5}$ concentrations during low and high
pollution (Fig. S6, Supplementary Material), though this association was only significant
during high pollution at Richmond (Table 2).
HRB frequency had a significant and positive influence on $PM_{2.5}$ only for the high
pollution condition at Chullora, Earlwood and Liverpool (Table 2 and Fig. 10). The
association between burn area and $PM_{2.5}$ during high pollution was significant at Liverpool
and Richmond only. The influence of Julian day on $PM_{2.5}$ showed significant non-linear,
increasing trends in all instances.
**4.3 Differences in covariate behaviours on HRB days with low versus high $PM_{2.5}$**
Having identified the most informative and consistent meteorological predictors using the
GAMMs, we compared their mean diurnal variation during the occurrence of HRBs and low
versus high $PM_{2.5}$ pollution:
*4.3.1 PBLH*
Taking Liverpool as an example, between 00:00 and 07:00 h during low pollution days when
HRBs have occurred, the PBLH is on average 100-200 m higher than during high pollution
days (Fig. 11; see Fig. S8-10 in the Supplementary Material for the other monitoring
stations). From late morning (ca. 10:00 h) until early evening (c. 19:00 h), the PBLH altitudes
of both $PM_{2.5}$ conditions are very similar, but after 19:00 h the PBLH is again higher during
low pollution.
*4.3.2 Total cloud cover*
During HRBs, mean diurnal variation of cloud cover is between 2 and 7 % greater during the
mornings and evenings of low pollution, compared to high pollution days (Fig. 11). In
contrast, there is minimal difference in cloud cover during the early afternoon of both
conditions.

### 4.3.3 Temperature

The temperature is 1 to 6 °C warmer between 00:00-08:00 h and 20:00-23:00 h during HRBs and low $PM_{2.5}$, in comparison to burns coinciding with high pollution (Fig. 11). However, there is a clear reversal in this trend from mid-morning to late afternoon during burns and high $PM_{2.5}$ when mean temperature is several degrees warmer than during HRBs and low pollution.

### 4.3.4 Wind speed

Mean diurnal wind speed is approximately 0.5 m s$^{-1}$ higher in the mornings and after 18:00 h during burns and low air pollution in comparison to speeds during high $PM_{2.5}$ (Fig. 11). In contrast, there is a minimal difference in wind speeds between 12:00 and 18:00 h.

## 5 Discussion

Air quality in Sydney is generally good. On the occasions when it is poor, atmospheric particulates are the principal cause, and HRBs are potentially one source of high particulate emissions. Sydney's population is projected to increase (~63 %) to over 8 million by 2061 (ABS, 2013), with much of the expansion occurring at the urban-bushland transition. Even if air quality remains stable, these demographic changes will increase exposure to particulate pollution. However, we observed increasing annual trends in $PM_{2.5}$ concentrations. In addition, projected decreases in future rainfall (Dai, 2013) and increases in fire danger weather are likely to increase fire activity and lengthen the fire season (Bradstock et al., 2014), thus amplifying fire-related particulate emissions. Changes in measurement instrumentation have a potential for introducing systematic biases in these annual $PM_{2.5}$ trends. Recently, based on the high correlation between beta attenuation monitors (BAMs), $PM_{2.5}$ measurements and long-term nephelometer visibility measurements at each monitoring site, the NSW Government (2016, 2017a, 2017b) reconstructed a more consistent annual average $PM_{2.5}$ time series. Their results also showed a tendency of increasing annual $PM_{2.5}$ levels near 2011/2012 in some Sydney subregions, as is consistent with the results from this study. Moreover, our study also indicates that the trends start increasing from 2011 during spring and winter, which pre-dates the instrumentation change. These results suggest that the instrumentation changes that occurred in 2012 are likely to have minimal impact the trend analysis reported in this analysis.

Relative to other pollutants such as $NO_x$ and $NO_2$, $PM_{2.5}$ concentrations are higher at weekends. $PM_{2.5}$ concentrations also start increasing in autumn with peaks in winter and spring. These patterns may reflect the timing of HRB occurrences, which occur mainly in autumn, spring and at weekends, though there is also increased domestic wood-fired heating during winter. Consequently, conducting multiple, concurrent HRBs during these periods might exacerbate $PM_{2.5}$ concentrations that are already high relative to baseline.

$PM_{2.5}$ concentrations tend to be dominated by organic matter (57%) during peak HRB periods in autumn. There is also contribution, in order of apportion, from elemental carbon, inorganic aerosol, and sea salt. This compares to summer months when sea salt plays a larger role, with organic matter making up just 34% (Cope et al. 2014). Other days where national $PM_{2.5}$ concentration standards have been exceeded have been attributed to

wildfires and dust storms. PM$_{2.5}$ concentrations also tend to be higher across the Sydney
basin during winter due to smoke from wood fire heaters used for residential heating,
however, exceedances of standards due to these emissions are rare (EPA, 2015).

**5.1 Primary covariates affecting PM$_{2.5}$ and how they differ during low and high pollution**

PBLH was the most consistent meteorological predictor of PM$_{2.5}$. It had a significant,
negative influence on PM$_{2.5}$ at all locations during HRBs and 'high pollution days'. There was
a marked difference in mean diurnal mixed layer heights between low and high pollution
conditions in the early morning (00:00-07:00 h) and from 20:00 to 23:00 h, with the PBLH
being approximately 100-200 m lower at these times during HRBs and high PM$_{2.5}$. During
these two time periods whilst the PBLH is low, mean cloud cover, temperature and wind
speeds are also lower relative to their magnitudes at corresponding times during low
pollution. Essentially, these early hours of cold, stable conditions with minimal turbulence
(i.e. conditions that are conducive to temperature inversions) prevent the dilution of PM$_{2.5}$.
These subdued conditions often coincide with the night time/early morning westerly cold
drainage flows and low mixing heights (inhibiting vertical dispersion), leading to the build-up
of PM$_{2.5}$ during mornings (Lu and Turco, 1995; Hart et al., 2006; Jiang et al., 2016b). These
pollution-conducive conditions are similar to those identified in Jiang et al. (2016a) as being
related to a ridge of high pressure extending across eastern Australia, resulting in light
north-westerly winds. These synoptically driven flows, although light, tend to enhance
nocturnal drainage flows, inhibit afternoon sea breeze formation, and allow the
transportation of pollutants across the Sydney basin to the coast. There is also a large
difference in mean diurnal temperatures between low and high pollution conditions from
late morning to early evening, with temperatures 3-4 °C warmer during high pollution.
During warmer daytime conditions, PM$_{2.5}$ can be potentially higher without fire events, for
instance, because these conditions tend to be coincident with increased precursor
emissions and generation of secondary organic aerosols in the air. Furthermore, the fact
that early morning and late evening temperatures tend to be lower during high pollution
conditions may indicate the presence of temperature inversions which hinder atmospheric
convection, leading to the collection of particulates that cannot be lifted from the surface.
Cold morning temperatures can also result in stronger drainage flows into the Sydney basin.
Consequently, if HRBs are being conducted during early mornings in the hills and mountains
to the west of Sydney, this could result in the dispersion of particles from such sources,
possibly into populated areas.
These findings indicate how the timing of HRBs can be altered to reduce their air
pollution impacts in Sydney. Conducting HRBs when the PBLH is forecast to be higher ought
to help reduce their air quality impacts in Sydney. More specifically, conducting HRBs later
in the morning (for example by a matter of hours) is one way of potentially reducing HRB air
quality impacts, because the PBLH generally starts increasing rapidly in height from 07:00
until 12:00 h. Fires conducted early in the morning when the PBLH is at its lowest, and
temperatures are cool will promote effects such as fire smoke residing near ground-level.
One constraint concerning later burn times is that wind speed typically increases as the day
progresses. However, the maximum mean diurnal wind speed was approximately 3 m s$^{-1}$
and occurred at 15:00 h. This is considerably lower than the RFS' upper-limit of 5.56 m s$^{-1}$ for
conducting safe HRBs (Plucinski and Cruz, 2015). An additional caution for conducting burns
later in the afternoon is that onshore coastal breezes can develop during afternoons. The
optimal timing of burns will also be dependent on other factors such as burn intensity,
lighting method, fuel/soil moisture and geographic location.
There was a negative association between cloud cover and PM$_{2.5}$ levels. It is possible
that fire agencies conduct fewer HRBs during cloudy conditions in case of rain. Rainfall (if
any) can also scavenge PM pollution out of the air. However, cloudless skies are also
associated with high pressure systems, and therefore cool air descending, resulting in a
stable calm atmosphere, and low PBLH that is not conducive to pollutant dispersion.
Although there were similarities in the influence of covariates between locations,
these associations often varied spatially. For example, mean diurnal PBLH and temperature
were lower at Richmond in the early morning and at night in comparison to the other
locations (Fig. S10, Supplementary Material). Richmond is further inland than the other
monitoring sites and is thus closer to the mountain range to the west of Sydney. The insights
gained into the spatial variation in the behaviour of covariates can support efforts to create
location-specific particulate pollution forecasts.
The north-westerly signal apparent for three of four locations during HRBs and high
pollution may reflect the fact that, overall, the majority of burns are conducted to the west,
north and north-west of Sydney (Fig. 2). From a management perspective, comparatively
greater attention might be devoted to adapting burn operations in these regions. In the case
of Richmond (where wind direction did not have a statistically significant influence), one
possible explanation is that the daily vector-averaging applied to the wind data has
smoothed out the signal associated with diurnal changes in wind directions (and speeds),
e.g. between drainage flow and sea breezes. Thus, to some degree, the signal of wind
influence may be suppressed in this case. Another contributing factor could be Richmond's
generally closer proximity to local burns. Also, its geographic location is quite different to
that of the other monitoring sites; it is further inland than the other sites and is thus closer
to the mountain range to the west of Sydney.
Using a different analysis approach, Price et al. (2012) found that the optimum
radius of influence of landscape fires on $PM_{2.5}$ was 100 km for Sydney. We found that whilst
close-proximity fires influenced air quality, fires up to approximately 300 km from
monitoring stations also potentially influenced $PM_{2.5}$. Longer-range exposures on regional
scales, particularly from multiple HRBs in an air-shed can impact communities at
considerable distance under certain atmospheric transport conditions (e.g. Liu et al., 2009).
Multiple concurrent burns are more likely to adversely affect air quality in Sydney, as
indicated by the significant, positive influence of the number of concurrent HRBs on $PM_{2.5}$
during high pollution days at all locations except Richmond. In general, greater numbers of
concurrent burns within a given air shed are likely to result in greater quantities of
particulate emissions. The area of these burns would also determine the amount of
particulate emissions generated. HRB total area per day was a statistically significant
predictor at two locations (Liverpool and Richmond). There are several possible
explanations for the fact that burn daily frequency and area are not significant predictors at
all locations. There will be some noise in total $PM_{2.5}$ concentrations contributed by other
emission sources, and this will vary with location. For example, Richmond differs from the
other monitoring sites in that it is near agricultural land, and so emission sources like soil
erosion and fertiliser use will introduce noise at this location. Investigating the relationships
between burnt area and fire-related tracer species to reduce the noise in total
concentrations contributed by other sources could be attempted in future work. There are
also uncertainties regarding how accurately the area actually burnt was recorded within
some polygons representing HRBs. In particular, to date it can be difficult to obtain timely
and accurate estimates of the actual area burnt. Moreover, larger burns are often further
away from the urban centres chosen, and are less frequent than smaller burns. In contrast,

moderate to small burns are more frequent and often scattered along the urban fringes (rather than confined to one location/direction) and thus have larger effect over the overall air quality within urban centres. Transport of smoke is also determined by interactions between basin terrains and local/synoptic wind conditions. However, the interaction between meso-scale geography and meteorological variables is a factor that could not be easily accounted for in the present study (i.e. each site is located in a different location, therefore each has differing topography and land use type).

## 6. Conclusions

Fine particulate concentrations are increasing in Sydney, and given projected increases in fire danger weather, intensification in fire activity is expected to further amplify fire-related $PM_{2.5}$ emissions. We identified the key meteorological factors linked to elevated $PM_{2.5}$ during HRBs. In particular, diurnal variation of the PBLH, cloud cover, temperature and wind speed have a pervasive influence on $PM_{2.5}$ concentrations, with these factors being more variable and higher in magnitude during the mornings and evenings of HRB days when $PM_{2.5}$ remains low. These findings indicate how the timing of HRBs can be altered to minimise pollution impacts. They can also support locality-specific forecasts of the air quality impacts of burns in Sydney and potentially other locations globally. In addition to mitigating wildfire risk, globally HRBs are used for forest management, farming, prairie restoration and greenhouse gas abatement. Future research should incorporate more sophisticated fire characteristics such as plume height and fuel moisture into analyses, and also consider the influence of climatic phenomena on particulate pollution. Synoptic features can also be incorporated into a future GAMM analysis, as well as modelling the diurnal evolution of $PM_{2.5}$ pollution due to HRB occurrences.

## Author contribution

G. Di Virgilio, M. A. Hart and N. Jiang conceived the research questions and aims. G. Di Virgilio designed and performed the analyses with contributions from all co-authors. G. Di Virgilio prepared the manuscript with contributions from all co-authors.

## Competing interests

The authors declare that they have no conflict of interest.

## Acknowledgments

This research was supported by the NSW Environmental Trust under grant 2014/RD/0147 and the NSW Office of Environment and Heritage (OEH). We thank OEH and the NSW Rural Fire Service NSW (NSW RFS) for providing the air quality, meteorological and fire data used in this research.

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

# Tables

**Table 1.** The area type, elevation, location, inter-annual (2005-2016) mean and standard deviation (SD) $PM_{2.5}$ concentration ($\mu g\ m^{-3}$) of each monitoring site.

| Site | Area Type | Elevation (m) | Lat | Lon | $PM_{2.5}$ Mean | $PM_{2.5}$ SD |
|------|-----------|---------------|-----|-----|-----------------|---------------|
| Chullora | Mixed residential/commercial | 10 | -33.89 | 151.05 | 7.56 | 4.13 |
| Earlwood | Residential | 7 | -33.92 | 151.13 | 7.26 | 4.34 |
| Liverpool | Mixed residential/commercial | 22 | -33.93 | 150.91 | 8.27 | 4.85 |
| Richmond | Residential/semi-rural | 21 | -33.62 | 150.75 | 6.85 | 6.29 |

**Table 2.** Adjusted $R^2$, $F$ and $p$-values for the smoothers of the optimal generalised additive mixed models (GAMM) applied to each monitoring site on days when hazard reduction burns occurred and with the data split into low and high air pollution conditions. Asterisks denote statistical significance: *** = $p < 0.001$; ** = $p < 0.01$; * = $p < 0.05$.

| Pollution Condition | Chullora | | Earlwood | | Liverpool | | Richmond | |
|---|---|---|---|---|---|---|---|---|
| | Low | High | Low | High | Low | High | Low | High |
| | $R^2$ | $R^2$ | $R^2$ | $R^2$ | $R^2$ | $R^2$ | $R^2$ | $R^2$ |
| | 0.38 | 0.44 | 0.29 | 0.60 | 0.39 | 0.60 | 0.29 | 0.47 |
| Variable | $F$ | $F$ | $F$ | $F$ | $F$ | $F$ | $F$ | $F$ |
| PBLH | 12.7 *** | 9.1 ** | 4.0 * | 13.2 *** | 3.3 * | 29.5 *** | 4.5 ** | 6.9 ** |
| MSLP | 0.0 | 0.4 | 0.0 | 3.7 | 0.0 | 2.0 | 0.0 | 1.6 |
| Temperature | 0.0 | 3.7 * | 0.8 | 2.9 | 4.6 * | 10.9 *** | 0.1 | 2.1 |
| Cloud cover | 12.9 *** | 16.9 *** | 9.2 ** | 9.9 *** | 10.6 ** | 16.9 *** | 2.9 | 7.6 ** |
| Rainfall | 2.0 | 1.6 | 5.7 * | 8.9 *** | 7.3 ** | 1.2 | 8.8 ** | 3.1 |
| Wind direction | 0.0 | 1.0 * | 0.0 | 1.7 ** | 0.0 | 2.5 *** | 0.0 | 0.2 |
| Wind speed | 0.1 | 2.4 | 3.4 | 3.9 ** | 1.0 | 0.0 | 5.8 * | 0.2 |
| HRBs daily frequency | 3.1 | 2.3 * | 0.0 | 2.8 * | 1.1 | 3.5 ** | 0.1 | 1.6 |
| HRBs area burnt daily | 6.8 *** | 1.4 | 3.0 | 0.3 | 1.6 | 5.7 ** | 1.2 | 9.5 *** |
| Julian Day | 12.1 *** | 5.9 *** | 10.1 *** | 10.7 *** | 18.8 *** | 11.9 *** | 32.3 *** | 2.6 |

**Figures**

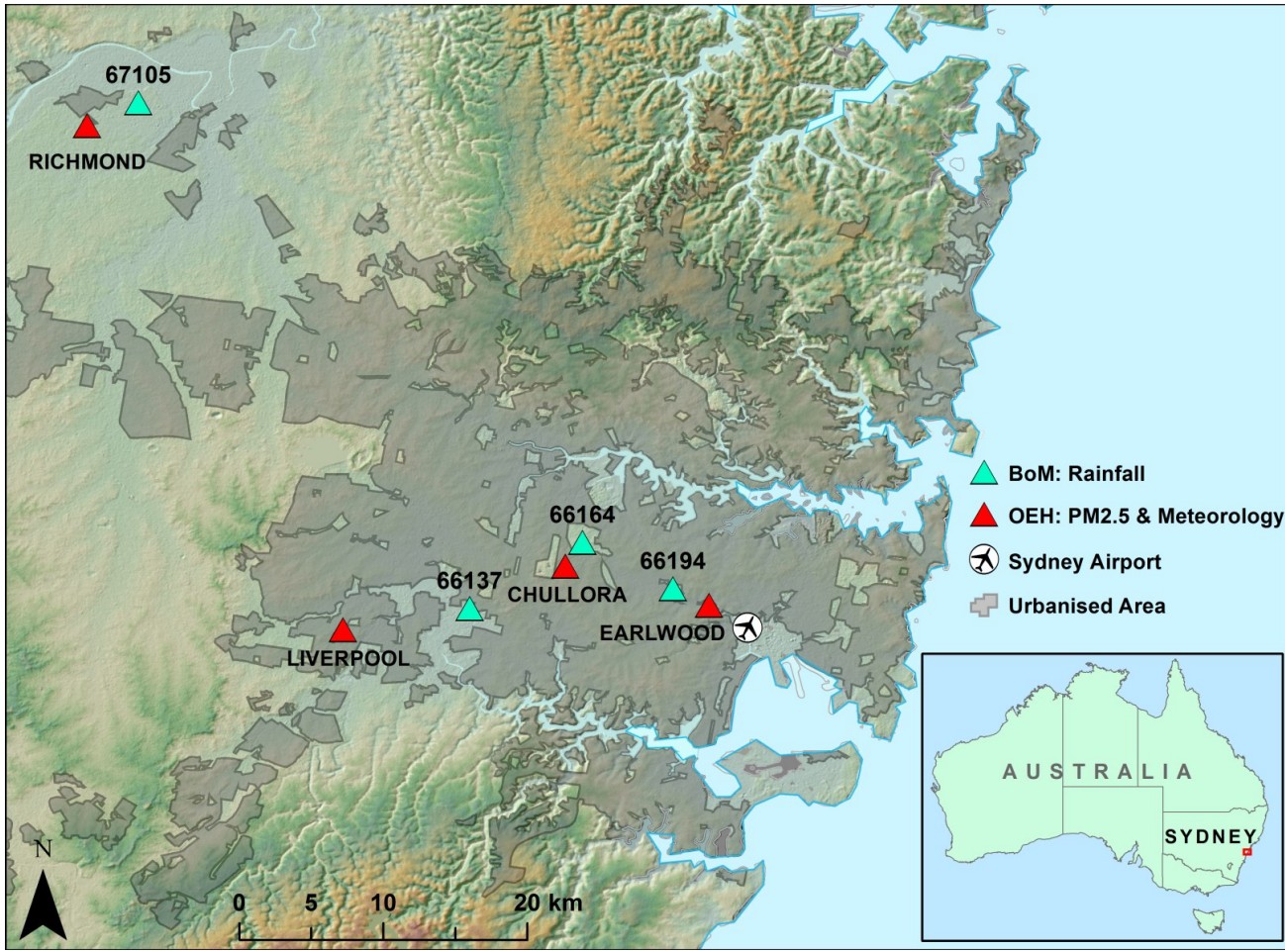

**Figure 1**. Locations of meteorological/PM$_{2.5}$ monitoring stations in the New South Wales Office of Environment and Heritage network in Sydney, Sydney Airport meteorological station, and Bureau of Meteorology (BoM) stations (with station numbers) from which rainfall data were obtained.

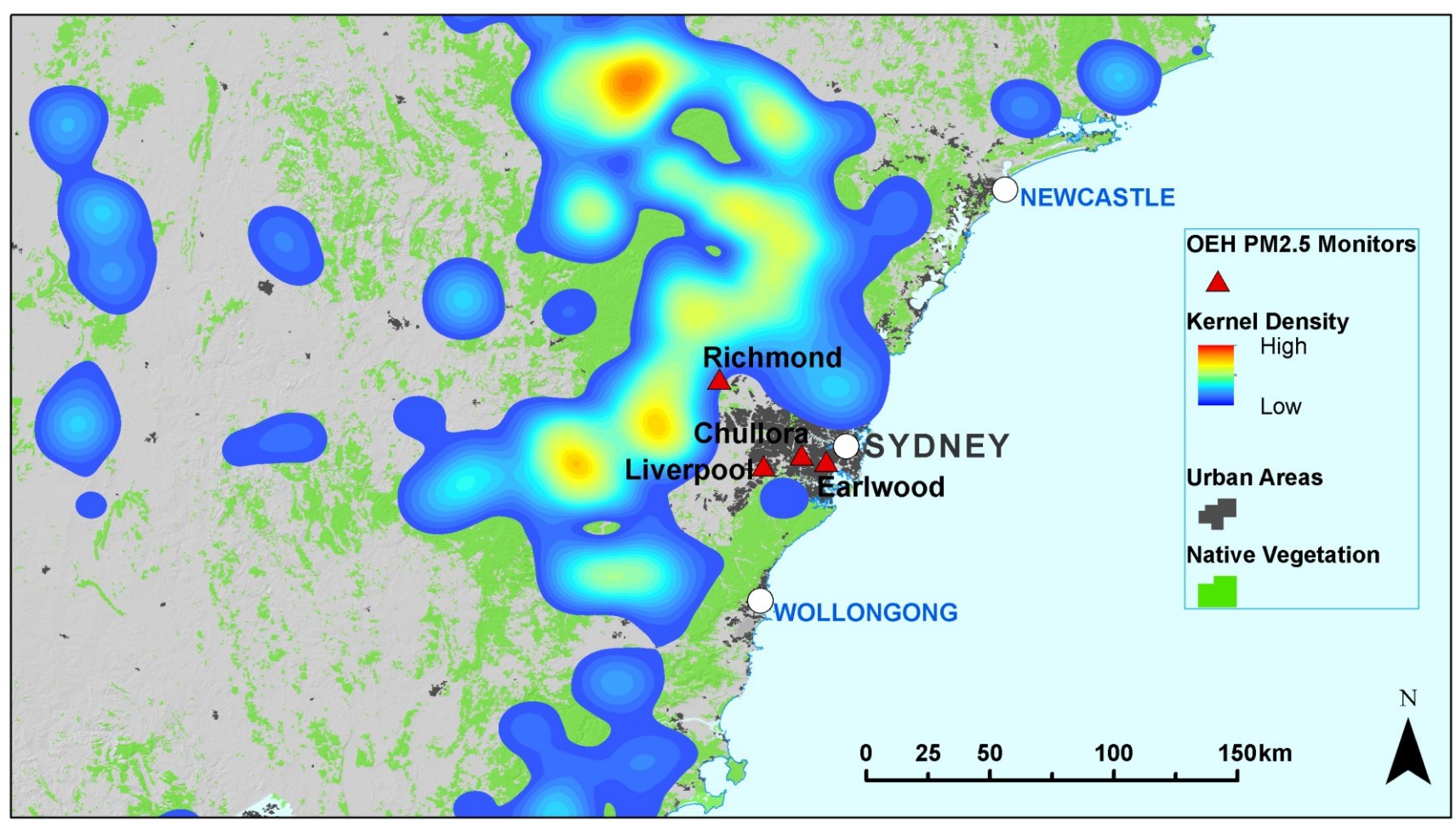

**Figure 2**. Kernel density function (magnitude-per-unit area) for hazard reduction burns (HRB) conducted in the vicinity of Greater Sydney (2004-2016). The warmer the colour of the kernel density surface, the more/larger HRBs that have occurred in that area. The kernel density calculation is weighted according to fire surface area.

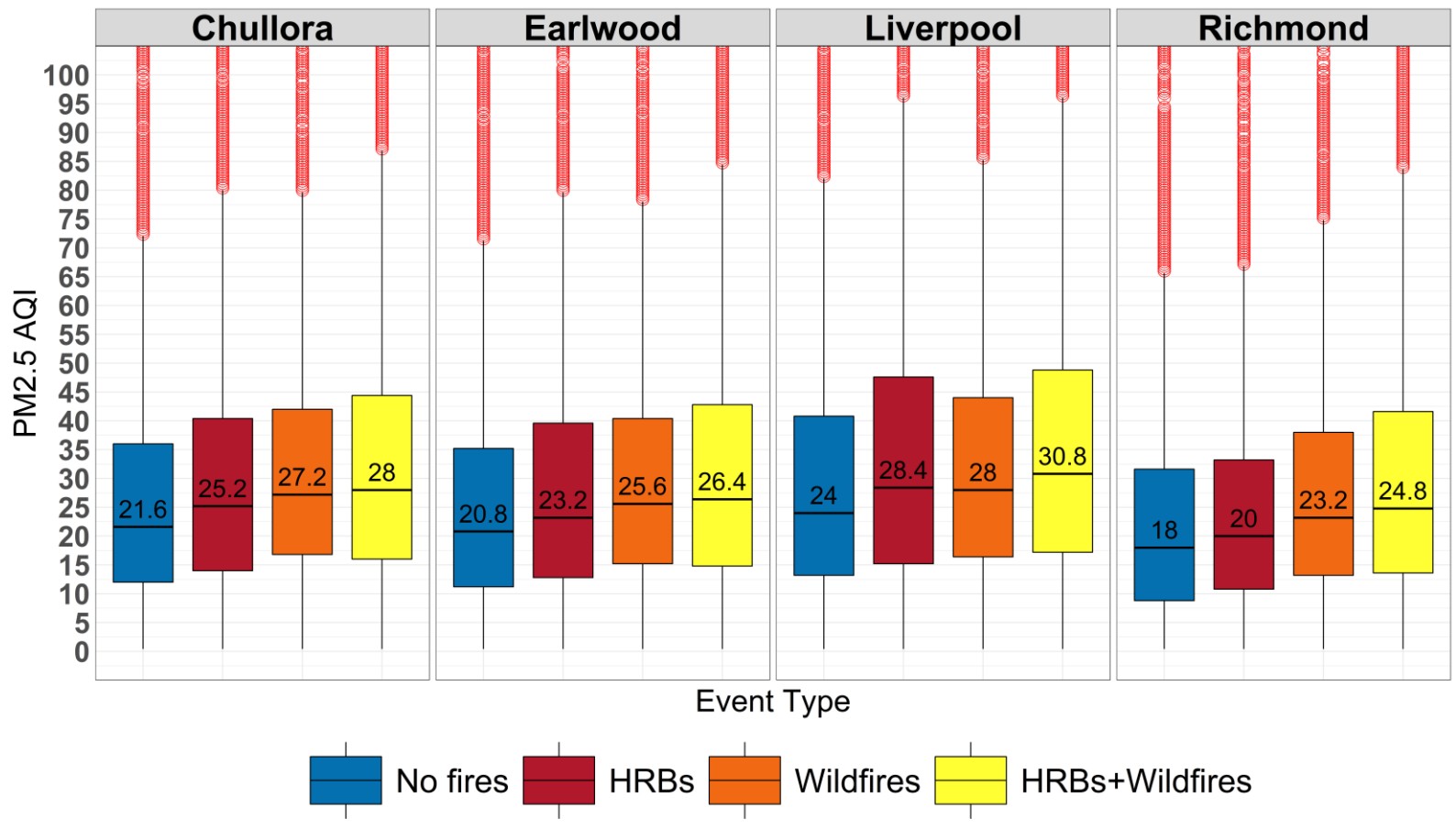

**Figure 3**. Boxplots showing the variation in PM$_{2.5}$ air quality index values (AQI) at four measurement sites in Sydney between 2011 and 2016 during days when there were no fires (neither hazard reduction burns (HRBs) or wildfires), days when only HRBs occurred without coincident wildfires, days when wildfires occurred without coincident HRBs, and days with concurrent HRBs and wildfires. Horizontal black lines on boxplots are median PM$_{2.5}$ AQIs and their corresponding values are shown above these lines. Red circles are outliers.

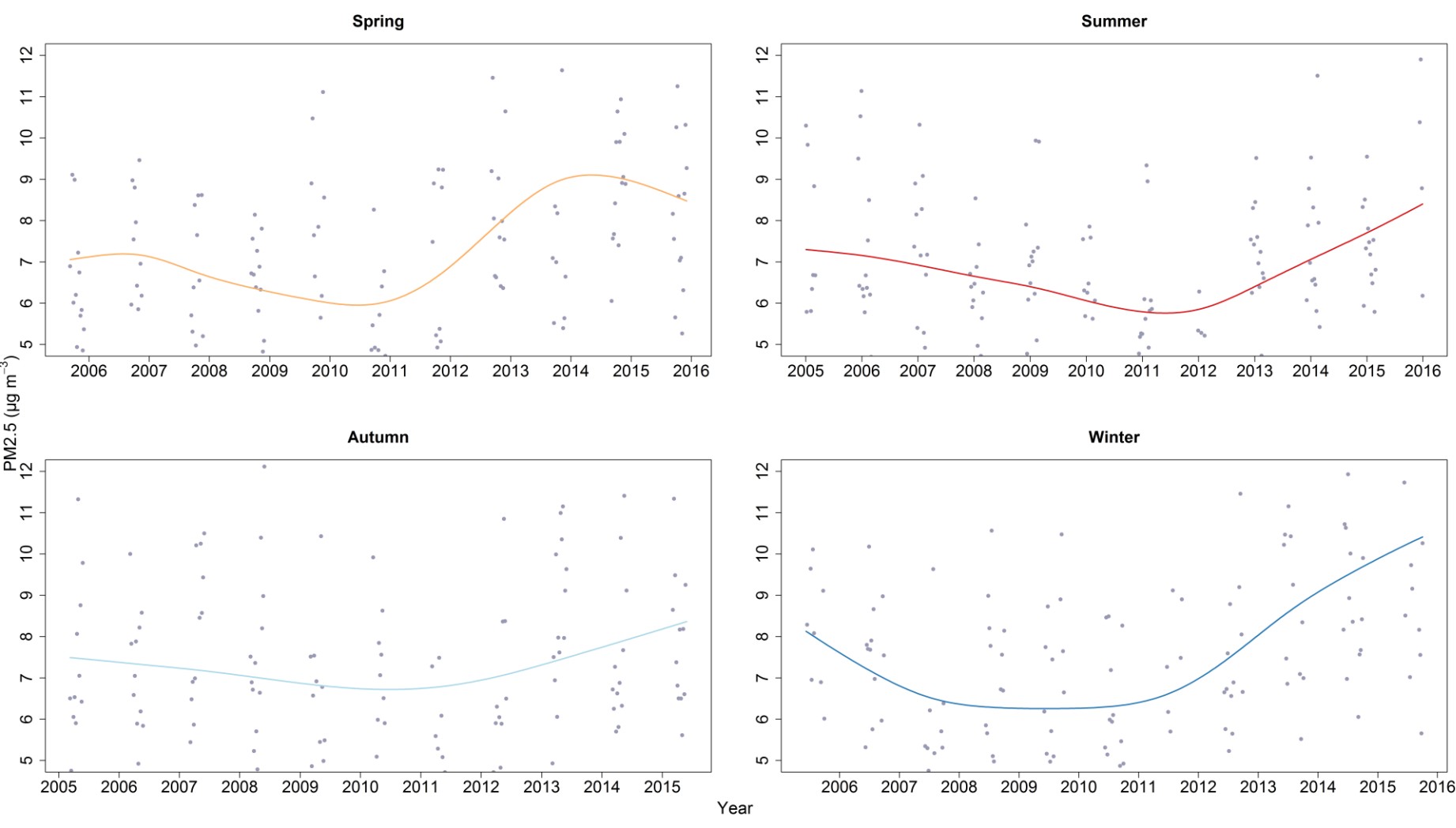

**Figure 4**. Annual trends in the weekly mean concentrations of $PM_{2.5}$ in Sydney, split by season for 2005 - 2015.

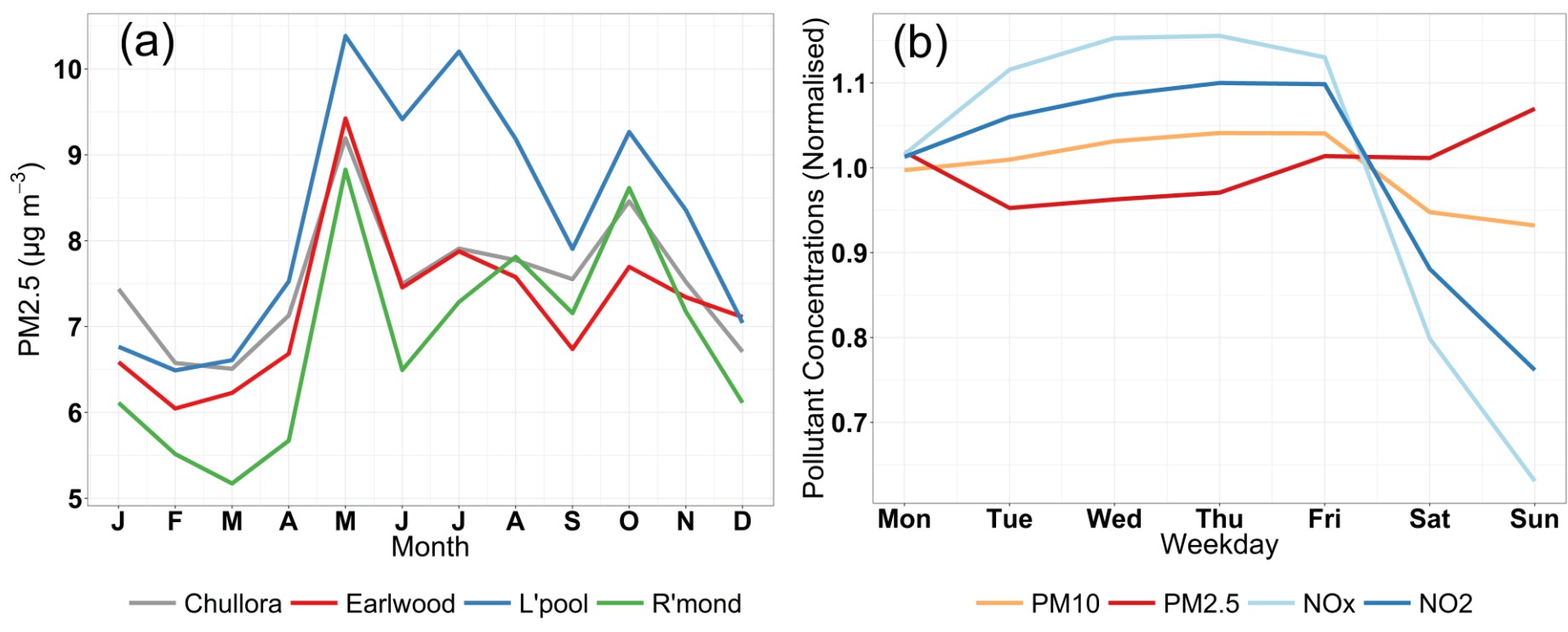

**Figure 5**. Mean monthly PM$_{2.5}$ concentrations for the period 2011 to August 2016 at four air quality monitoring sites in Greater Sydney (a). Southern hemisphere seasons are summer (DJF), autumn (MAM), winter (JJA) and spring (SON). Mean daily normalised concentrations of PM$_{2.5}$ compared to the variations of PM$_{10}$, NO$_2$ and NO$_x$ (b).

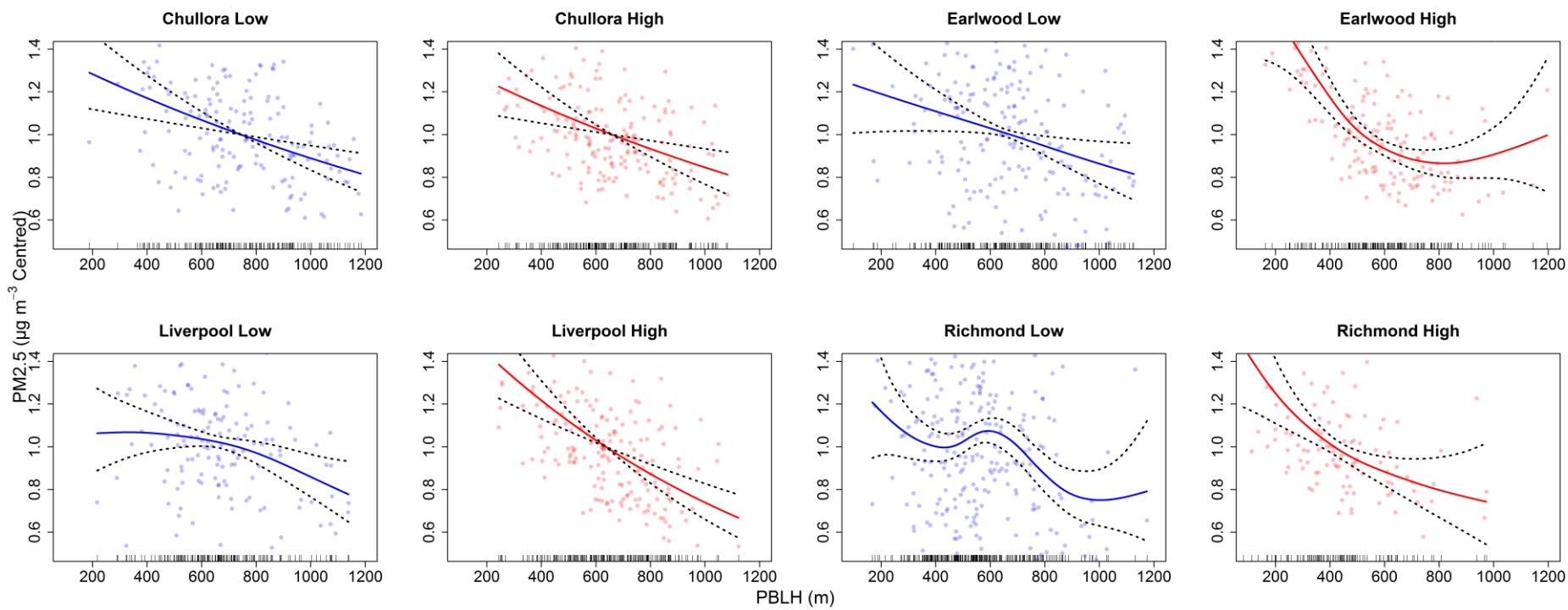

**Figure 6**. The contribution by the planetary boundary layer height (PBLH) component of the generalised additive mixed model (GAMM) linear predictor to fitted PM$_{2.5}$ values (μg m$^{-3}$, centred). The solid lines are the fitted curves. Dotted lines are 95% confidence bands. Dots are partial residuals.

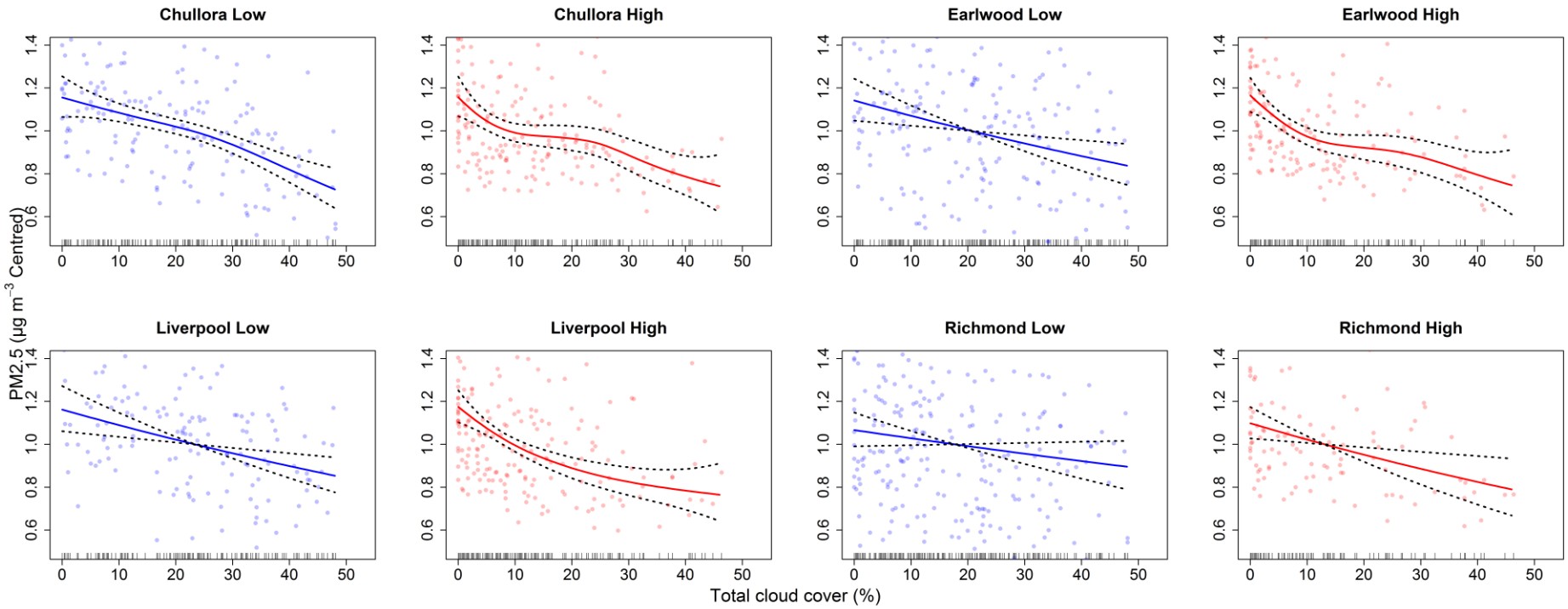

**Figure 7**. The contribution by the cloud cover component of the GAMM linear predictor to fitted $PM_{2.5}$ values ($\mu$g m$^{-3}$, centred). The solid lines are the fitted curves. Dotted lines are 95% confidence bands. Dots are partial residuals.

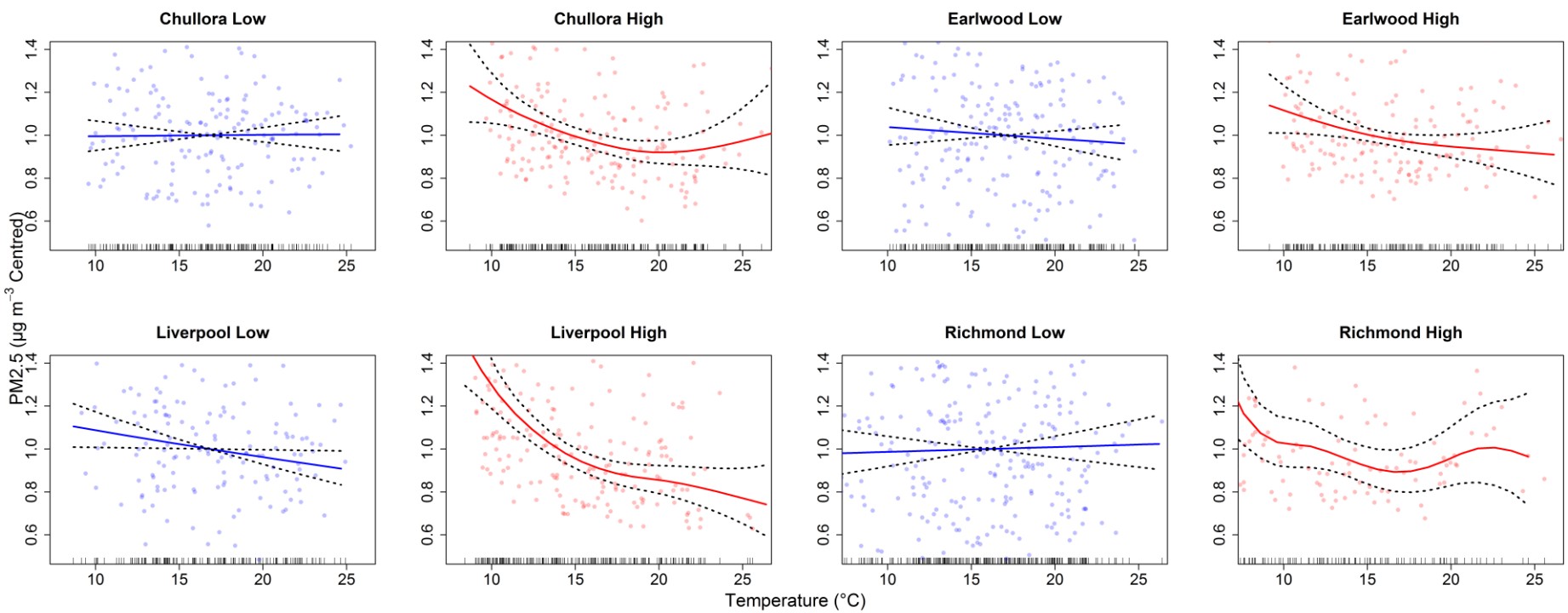

**Figure 8**. The contribution by the temperature component of the GAMM linear predictor to fitted PM$_{2.5}$ values (μg m$^{-3}$, centred).

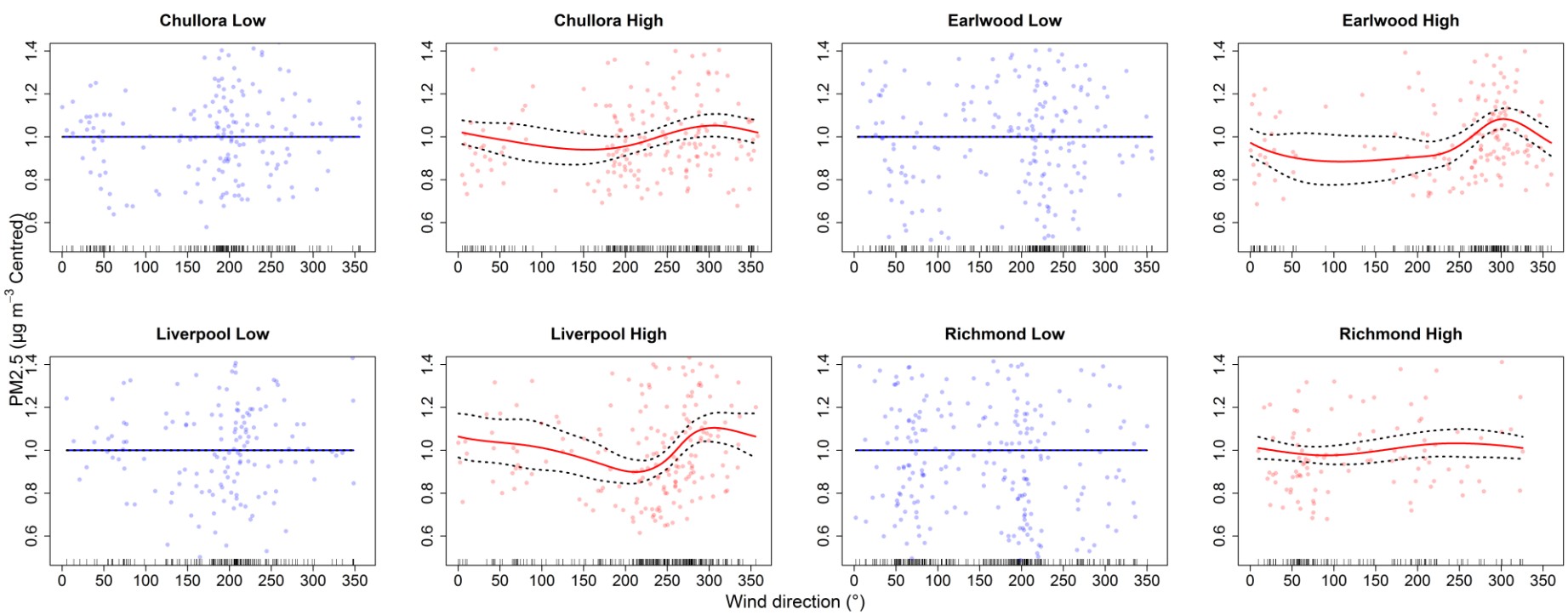

**Figure 9**. The contribution by the wind direction component of the GAMM linear predictor to fitted PM$_{2.5}$ values (µg m$^{-3}$, centred).

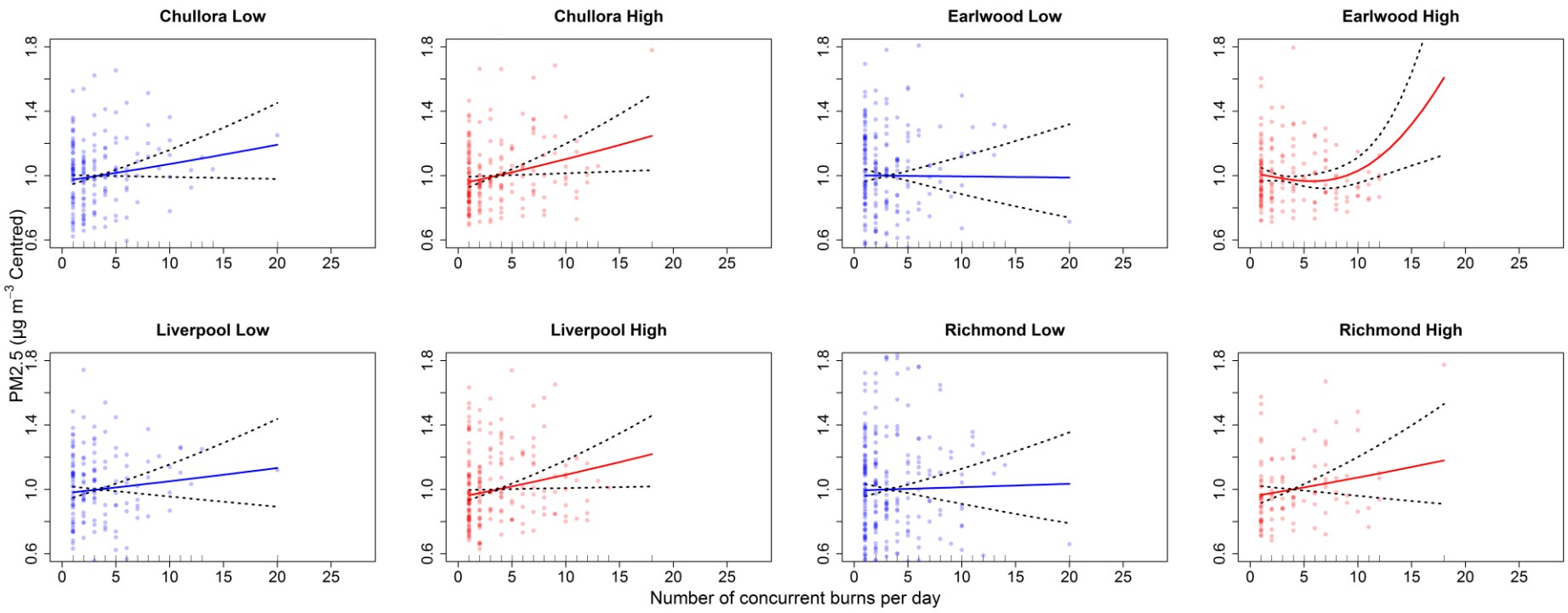

**Figure 10**. The contribution by the hazard reduction burn (HRB) daily frequency (number of concurrent burns per day) component of the GAMM linear predictor to fitted PM$_{2.5}$ values (µg m$^{-3}$, centred).

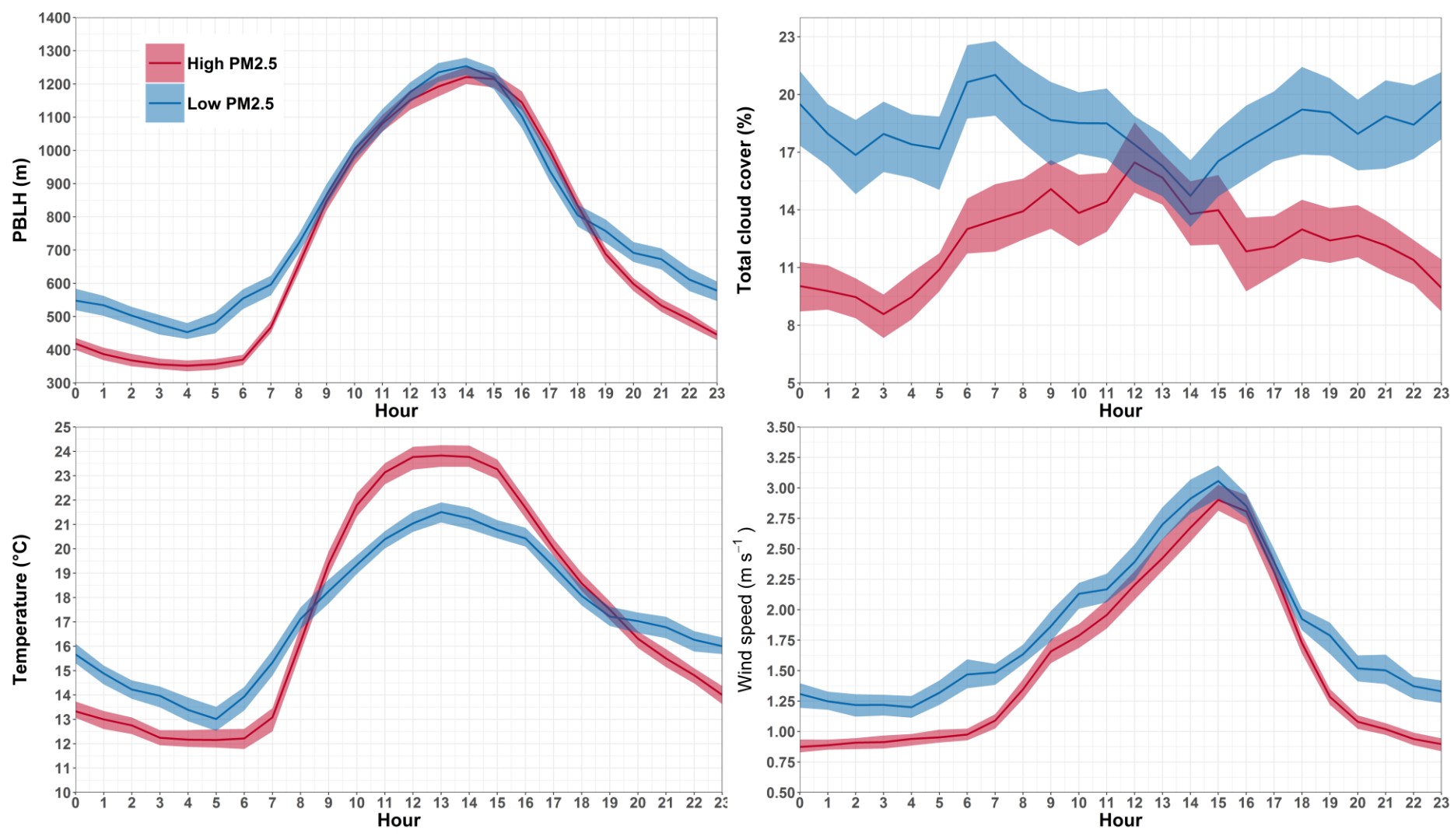

**Figure 11**. Mean diurnal variation of hourly PBLH, total cloud cover, temperature and wind speed for low versus high $PM_{2.5}$ pollution during HRBs at Liverpool, Sydney (see Figures S8-S10 for other stations). Shading represents the 95 % confidence intervals of the means.