# Peer review of "Meteorological controls on atmospheric particulate pollution during hazard reduction burns"

_Atmospheric Chemistry and Physics, 2017_

## Referee Comment (RC1) · Anonymous Referee #2 · 27 Oct 2017

General comments:

Fires are growing threats to ecosystems and human society in many regions along with climate change. Accordingly, hazard reduction burns (HRBs) or prescribed burns as discussed in this work have gained more attention since they are effective techniques in wildland and forest management for large wildfire prevention. It's important to conduct HRBs skillfully and safely to ensure they are more controllable and less annoying with mitigated negative impacts on air quality and public health. Virgilio et al. in this study used generalized additive mixed models (GAMMs) to examine the meteorological impact on atmospheric particulate pollution during HRBs in Sydney, Australia,
which could benefit HRBs practices in Sydney and other similar regions. The GAMM model generation and selection framework is suitable for this air pollution meteorology study to explicitly take account of collinearity and autocorrelation problems in the predictor and response variables. The manuscript is well-organized and -written, and the conclusion is clear and concise. However, more comprehensive analysis with physical interpretation and implications of modeling results should be added to increase its merits to the fire risk management community. Therefore, I would suggest its publication on the discussion forum of the ACP journal after addressing the specific comments listed below.

Specific comments:

(1) In Table 2 for GAMM model selection, the significant predictor variable groups are quite different among monitoring sites and pollution conditions. I would expect wind direction to be an important factor for local high air pollution during HRBs, but I only found it significant for two sites (Earlwood and Liverpool) rather than the others (Chullora and Richmond). Also the HRBs daily frequency is more significant in the high pollution condition than in the low pollution condition, while HRBs daily burnt area is opposite in general. How to explain these differences and what is the implication for statistical model selection and HRB implementation?

(2) Another question about the relations between meteorological variables and PM2.5 concentrations is the understanding and interpretation of these statistical findings. The authors suggested that "PBLH and total cloud cover were the most consistent predictors of elevated PM2.5 during HRBs" (line 236-237). It is relatively easy to understand the relation between PBLH and PM2.5 concentrations, while it is not that intuitive to interpret the connection between cloud cover and PM2.5. My guess is people tend to conduct less HRBs in cloudy days in case of rain, which might explain the negative influence of cloud cover on predicted PM2.5. The authors should check the original datasets with more detailed explanation of these relations.

(3) I have some concerns with the concentration data used for the trend analysis. It is noted in section 2.2 that the PM2.5 measurement instrument changed since 2012, which might introduce systematic biases in the annual trends in Fig.4. The authors should be cautious about the increasing trends in PM2.5 concentrations after 2011 in Fig.4 and discuss more in the main text about this problem and potential needs for bias correction.

(4) In Fig.11, the fitting seems to be dominated by very few extreme large value samples, especially in the high pollution group. How robust are these relations?

(5) The authors suggested a maximum spatial distance of approximately 300km for the HRB influence on air quality, which is much larger than similar studies of about 100km. Why? What is the temporal scale of HRB influence? How many days after HRBs would the influence on air quality be negligible?

(6) Usually burnt area is one of predominant factors for fire emissions that affect air quality directly. It's a bit surprising that HRB total burnt area per day is not an effective predictor in this study. Though the authors attributed this result to the uncertainty in the burnt area estimation, it's still not very convincing. Probably the authors could examine the correlation between burnt area and fire related tracer species instead of total PM2.5 concentrations to reduce noise in total concentrations contributed by other emission sources.

(7) It's suggested that the authors conduct more comprehensive analysis of air pollution meteorology from other perspectives than individual meteorological variables. For instance, composite analysis on synoptic weather patterns in addition to these variables might be helpful to understand the impact of meteorological conditions on air pollution. The identification of critical weather patterns also benefits the interpretation of statistical relations between meteorological variables and ambient air pollution.

Technical corrections:

(1) The locations of monitoring stations such as Earlwood and Chullora are different in Fig.1 and Fig.2. Please double check the location for each site in these figures.

(2) The subplot titles and axis labels are too small in Figs.4, 6-11. The legends are also missing in these figures.

(3) Please clarify the calendar months for each season in the Southern Hemisphere in line 139-140 or show the seasons in Australia in Fig.5(a) for clear interpretation.

(4) Please indicate the resampling number of the bootstrap method in line 224.

---

## Referee Comment (RC2) · Anonymous Referee #3 · 11 Dec 2017

This paper use the a generalized additive mixed modelling (GAMM) method to find the 9 out of 16 selected meteorological and date variables, e.g. PBLH, total cloud cover, wind speed and temperature etc, which have more influence to PM2.5 variation induced by hazard reduction burns (HRB) in Sydney, Australia. The paper has more technique part than the scientific significance although it has clear method description and organized well. However, my concern is that some result figures can not support enough on the conclusion about the relationship between PM2.5 emission and meteorological variables. For example, the Chullora Low and Earlwood Low in figure 5 show very scattered partial residuals, even though the fitted curves can be conducted anyway and show trends, but the confidence on these trends is kind of low. The similar

scattered PM2.5 plots are found in figures 6-13 as well. I wonder if other function selected for GAMM can make better simulations, please see more detail in my question 2.

More questions are listed following: 1, at line 97, author includes many meteorological variables in analysis. Does any influence come from ocean variables? such as sea surface temperature, ENSO etc.

2, at line 155, author choose an identity link function in GAMM. However a log link function seem more suitable for the nonlinear relationship between air quality and meteorological variables.

3, at line 169, is lag in days or months in Figure S1?

4, at line 188, is median value of PM2.5 calculated by HRB days or the whole year? I prefer to use all days instead of only HRB days.

5, at line 199, how is definition of a threshold of 3.5 here?

6, at line 204 and 206, why is 300km a maximum burn distance instead of 500km?

7, at line 220, figure 3 show increasing trends in all season after 2011. Author also mentioned that the new instruments had been used to measure PM2.5 concentration since 2012. Does the instrument change contribute the increasing trend?

8, at line 223, figure 4a show PM2.5 is lower in Spring than other seasons, but figure 3 does now show PM2.5 lower in Spring. Any explanation?

9, at line 224, what reason is the PM2.5 higher in weekend while other pulltants get lower?

---

## Author Comment (AC1) · 22 Jan 2018

**acp-2017-491: Meteorological controls on atmospheric particulate pollution during hazard reduction burns**

Dear Dr Qian,

Thank you for the opportunity to submit a revised version of our manuscript in light of reviewer comments from the public discussion.

We thank the two reviewers for their valuable and constructive input. As you will see from our detailed point-by-point responses below, we have carefully gone through all of the reviewer comments and suggestions.

We believe the reviewer comments have strengthened the manuscript and hope you find the revisions satisfactory. We are, of course, happy to make further changes if necessary and look forward to a decision on our revised manuscript in due course.

Kind regards,

Giovanni Di Virgilio, Melissa Hart, and Ningbo Jiang

**Table 1. Responses to comments by anonymous referee #2**

| | Comment | Response |
|---|---|---|
| 1 | In Table 2 for GAMM model selection, the significant predictor variable groups are quite different among monitoring sites and pollution conditions. I would expect wind direction to be an important factor for local high air pollution during HRBs, but I only found it significant for two sites (Earlwood and Liverpool) rather than the others (Chullora and Richmond). | In light of feedback from reviewers and colleagues, we amended the GAMM analyses to reflect the following changes:

- Removed the very rare occurrences when there were days with an extremely high number of concurrent burns (i.e. we removed the few instances when there were 25 concurrent burns), which as reviewer 2 noted in comment #5 below seemed to be dominating fitting.

- Used a log link function instead of an identity link; using the former led to a general improvement in model performance.

- Removed day of year as a predictor, as it was not a significant covariate; hence the study was focused on common effects across years.

The results generated using the amended GAMMs were generally consistent with the previous GAMM results. However, wind direction now has a statistically significant relationship with PM2.5 concentrations in the high pollution condition in 3 of 4 locations (see Table 2 and Figure 9 in the revised manuscript). In the case of Richmond (where wind direction does not have a statistically significant influence), one possible explanation is that the daily vector-averaging applied to the wind data has smoothed out the signal associated with diurnal changes in wind directions (and speeds), e.g. between drainage flow and sea breezes. Thus, to some degree, the signal of wind influence may be suppressed in this case. Another contributing factor could be Richmond's generally closer proximity to local burns. Its geographic location is quite different to that of the other sites studied; it is further inland than the other monitoring sites and is thus closer to the mountain range to the west of Sydney. |

| | | The above points are stated at pp. 15-16 lines 392-400 in the revised manuscript. |
|---|---|---|
| 2 | Also the HRBs daily frequency is more significant in the high pollution condition than in the low pollution condition, while HRBs daily burnt area is opposite in general. How to explain these differences and what is the implication for statistical model selection and HRB implementation? | The results of the amended GAMMs still show that HRB daily frequency is generally significant in the high pollution condition. Daily burnt area is now also significant in the high pollution case for Liverpool and Richmond.

There are several possible explanations for the fact that burn daily frequency and area are not significant predictors in all cases. There will be some noise in total PM2.5 concentrations contributed by other emission sources. For instance, Richmond is near agricultural land and so emission sources like soil erosion and fertiliser use will introduce noise at this location. Moreover, larger burns are often further away from the urban centres chosen, and are less frequent than smaller burns. In contrast, moderate to small burns are often more frequent and scattered along the urban fringes (rather than confined to one location/direction) and thus have larger effect over the overall air quality within urban centres. Transport of smoke is also determined by interactions between basin terrains and local/synoptic wind conditions. The interaction between meso-scale geography and meteorological variables is a factor that could not be easily accounted for in the present study (i.e. each site is located in a different location/therefore different topography and land use type).

Therefore, the following text has been added to the Discussion at pp. 16-17, lines 413-431. Please note that the text in blue was already present in the previous submission:

"There are several possible explanations for the fact that burn daily frequency and area are not significant predictors at all locations. There will be some noise in total PM$_{2.5}$ concentrations contributed by other emission sources, and this will vary with location. For example, Richmond differs from the other monitoring sites in that it is |

| | | near agricultural land, and so emission sources like soil erosion and fertiliser use will introduce noise at this location. Investigating the relationships between burnt area and fire-related tracer species to reduce the noise in total concentrations contributed by other sources could be attempted in future work. There are also uncertainties regarding how accurately the area actually burnt was recorded within some polygons representing HRBs. In particular, to date it can be difficult to obtain timely and accurate estimates of the actual area burnt. Moreover, larger burns are often further away from the urban centres chosen, and are less frequent than smaller burns. In contrast, moderate to small burns are often more frequent and scattered along the urban fringes (rather than confined to one location/direction) and thus have larger effect over the overall air quality within urban centres. Transport of smoke is also determined by interactions between basin terrains and local/synoptic wind conditions. However, the interaction between meso-scale geography and meteorological variables is a factor that could not be easily accounted for in the present study (i.e. each site is located in a different location, therefore each has differing topography and land use type)." |
|---|---|---|
| 3 | Another question about the relations between meteorological variables and PM2.5 concentrations is the understanding and interpretation of these statistical findings. The authors suggested that "PBLH and total cloud cover were the most consistent predictors of elevated PM2.5 during HRBs" (line 236-237). It is relatively easy to understand the relation between PBLH and PM2.5 concentrations, while it is not that intuitive to interpret the connection between cloud cover and PM2.5. My guess is people tend to conduct less HRBs in cloudy days in case of rain, which might explain the negative influence of cloud cover on predicted PM2.5. The authors should check the original datasets | An additional explanation for the negative association between cloud cover and PM2.5 concentrations is that cloudless skies are associated with high pressure systems, and therefore cool air descending, resulting in a stable calm atmosphere, and low PBLH, that is not conducive to pollutant dispersion.

Therefore, the following text has been added in the discussion that mentions the suggested explanation by the reviewer, and also the above:

"There was a negative association between cloud cover and PM2.5 levels. It is possible that fire agencies conduct fewer HRBs during cloudy conditions in case of rain. Rainfall (if any) can also scavenge PM pollution out of the air. However, cloudless skies are also associated with high pressure systems, and therefore cool air descending, resulting in a stable calm atmosphere, and low PBLH, that is not conducive to pollutant dispersion." (p. 15, lines 377-81) |

| | | |
|---|---|---|
| | with more detailed explanation of these relations. | |
| 4 | I have some concerns with the concentration data used for the trend analysis. It is noted in section 2.2 that the PM2.5 measurement instrument changed since 2012, which might introduce systematic biases in the annual trends in Fig.4. The authors should be cautious about the increasing trends in PM2.5 concentrations after 2011 in Fig.4 and discuss more in the main text about this problem and potential needs for bias correction. | Changes in measurement instrumentation have a potential for introducing systematic biases in these annual PM2.5 trends. Recently, based on the high correlation between beta attenuation monitors (BAMs), PM2.5 measurements and long-term nephelometer visibility measurements at each monitoring site, the NSW Government (2016, 2017a, 2017b) reconstructed a more consistent annual average PM2.5 time series. Their results also showed a tendency of increasing annual PM2.5 levels near 2011/2012 in some Sydney subregions, as is consistent with the results from this study. Moreover, our study also indicates that the trends start increasing from 2011 during spring and winter, which pre-dates the instrumentation change. These results suggest that the instrumentation changes that occurred in 2012 are likely to have minimal impact the trend analysis reported in this analysis.

We have expanded the first paragraph of the Discussion to raise these points and three studies conducted by the NSW government investigating this issue are cited (please see p.13, lines 308-318 in the main text; also p.7 in 'Air Quality in NSW' by the NSW Government , 2017a). |
| 5 | In Fig.11, the fitting seems to be dominated by very few extreme large value samples, especially in the high pollution group. How robust are these relations? | Prior to revising the GAMM analyses, as stated above (please see comment #1) the data included a few rare occurrences when there were ~25 concurrent burns. In case these instances were dominating the fitting, we removed them and re-ran the analyses. The new results are generally consistent with the results in the initial submission, with the changes noted above in #1 and #2. |
| 6 | The authors suggested a maximum spatial distance of approximately 300km for the HRB influence on air quality, which is much larger than similar studies of about 100km. Why? What is the temporal scale of HRB | An example previous study (Price et al. 2012; PLoS One) found that fire hotspots (both wildfires and HRBs) within 100 km of Sydney and 400 km of Perth influenced pollution levels. However, these authors used a different approach to the one used here, in that they used generalised linear models of the relationship between fire radiative power |

| | influence? How many days after HRBs would the influence on air quality be negligible? | (FRP) and PM2.5 concentration.

The primary focus of this study was on how meteorological variables influence PM2.5 concentrations during HRBs. Examining the temporal scale of HRB influence is a good suggestion that we are implementing in a separate study using a modelling/simulation approach with the weather research and forecasting (WRF) model. |
|---|---|---|
| 7 | Usually burnt area is one of predominant factors for fire emissions that affect air quality directly. It's a bit surprising that HRB total burnt area per day is not an effective predictor in this study. Though the authors attributed this result to the uncertainty in the burnt area estimation, it's still not very convincing. Probably the authors could examine the correlation between burnt area and fire related tracer species instead of total PM2.5 concentrations to reduce noise in total concentrations contributed by other emission sources. | Since revising the GAMM analyses, burnt area is a statistically significant predictor of PM2.5 levels in several instances. Where it is not, we concur with the reviewer that factors such as noise introduced by other emissions sources such as soil erosion, agriculture and domestic wood-fired heating could be introducing noise. Investigating the relationships between burnt area and fire-related tracer species (instead of PM2.5 concentrations) to reduce the noise in total concentrations contributed by other sources is a good suggestion that can be attempted in future work. We now mention the confounding effects of noise, as well as other issues in the Discussion at pp. 16-17, lines 415-431. |
| 8 | It's suggested that the authors conduct more comprehensive analysis of air pollution meteorology from other perspectives than individual meteorological variables. For instance, composite analysis on synoptic weather patterns in addition to these variables might be helpful to understand the impact of meteorological conditions on air pollution. The identification of critical weather patterns also benefits the interpretation of statistical relations between meteorological variables and ambient air pollution. | This is a good suggestion that we are implementing in further research that extends initial work conducted by Jiang et al. (2016). The scope of the current investigation was focused on local-scale meteorological predictors. We are conducting a similar study as in Jiang et al. (2014): Jiang, Ningbo & Dirks, Kim & Luo, Kehui. (2014) where we use a WRF modelling approach incorporating a larger suite of simulated variables including oceanographic (e.g. ENSO, SST) and synoptic-scale meteorology over a longer time scale. |
| | **Technical corrections:** | |

| i | The locations of monitoring stations such as Earlwood and Chullora are different in Fig.1 and Fig.2. Please double check the location for each site in these figures. | The locations of monitoring stations are now correct in Figures 1 and 2. |
|---|---|---|
| ii | The subplot titles and axis labels are too small in Figs.4, 6-11. The legends are also missing in these figures. | Subplot title and axis labels have been made larger on Figure 4, and Figures 6-11. |
| iii | Please clarify the calendar months for each season in the Southern Hemisphere in line 139-140 or show the seasons in Australia in Fig.5(a) for clear interpretation. | The main text now clarifies the calendar months for the seasons in the main text and also in the caption for Figure 5a. |
| iv | Please indicate the resampling number of the bootstrap method in line 224. | The number of bootstrap replicates used was 1000 - this is now stated at lines 226-8, p 9 in the revised manuscript. |

**Table 2. Responses to comments by anonymous referee #3**

| # | Comment | Response |
|---|---|---|
| 1 | The paper has more technique part than the scientific significance although it has clear method description and organized well. However, my concern is that some result figures cannot support enough on the conclusion about the relationship between PM2.5 emission and meteorological variables. For example, the Chullora Low and Earlwood Low in figure 5 show very scattered partial residuals, even though the fitted curves can be conducted anyway and show trends, but the confidence on these trends is kind of low. | Please note that Figure 5 referred to by the referee is Figure 6 in the revised manuscript.

The models were re-run after incorporating the suggestions by this reviewer (e.g. to use a log link function) and also some suggestions made by reviewer #2 above. These changes were as follows:

- Removed the very rare occurrences when there were days with an extremely high number of concurrent burns (i.e. we removed the few instances when there were 25 concurrent burns), which as reviewer above noted in Table 1, comment #5 seemed to be dominating fitting. |

|   |   |   |
|---|---|---|
|   |   | - Used a log link function instead of an identity link; using the former led to a general improvement in model performance.

- Removed day of year as a predictor, as it was not a significant covariate.

The results generated using the amended GAMMs were generally consistent with the previous GAMM results. However, for several variables (such as the case for PBLH referred to by the reviewer; see Figure 6), the confidence limits (zone of dashed lines) are slightly narrower than was previously the case. Also, the confidence limits are generally only wide/broad at extreme values, which could be expected.

The scattering nature of partial residuals reflect to some degree the complexity of the problem, i.e. there are multiple factors influencing the variability of PM2.5 levels in Sydney. The partial residuals were shown in these figures just to give an indication of the scale of variability in the data. For a well-fitting model the partial residuals should be evenly scattered around the curve to which they relate, and for most variables this is the case. |
| 2 | I wonder if other function selected for GAMM can make better simulations, please see more detail in my question [#4 below]. | We have implemented the reviewer's suggestion – please see our response at point #4 below. |
| 3 | More questions are listed following: 1, at line 97, author includes many meteorological variables in analysis. Does any influence come from ocean variables? such as sea surface temperature, ENSO etc. | In the present study, we did not include ENSO, as our observational data set covers a ~5.5-year timespan, whereas ENSO has a cycle of 2-7 years, hence the data set is not long enough to analyse its effects. However, to include such oceanographic variables is a good suggestion. Such a study is planned and will be undertaken in a separate analysis, as is similar to Jiang et al. (2014) but using the GAMM method for NSW.

We are also conducting a similar study as in Jiang et al. (2014): Jiang, Ningbo & Dirks, Kim & Luo, Kehui. (2014) where we use a weather research and forecasting (WRF) modelling approach incorporating a larger suite of simulated variables including |

| | | oceanographic (e.g. ENSO, SST) and synoptic-scale meteorology over a longer time scale at 5 km spatial resolution. |
|---|---|---|
| 4 | At line 155, author choose an identity link function in GAMM. However a log link function seem more suitable for the nonlinear relationship between air quality and meteorological variables. | We re-ran the GAMM analyses using a log link function, which generally improved model performance/fit relative to using an identity link. |
| 5 | at line 169, is lag in days or months in Figure S1? | Lag is in days; the supplementary material now states this in the captions for Figures S1-2. |
| 6 | at line 188, is median value of PM2.5 calculated by HRB days or the whole year? I prefer to use all days instead of only HRB days. | Correct: the calculation of the median value of PM2.5 used all days, not just HRB days. |
| 7 | at line 199, how is definition of a threshold of 3.5 here? | The variance inflation factor (VIF) threshold of 3.5 was selected as a compromise between the thresholds of 10 and 3 stated in Zuur et al. (2010) 'A protocol for data exploration to avoid common statistical problems'.

The main text has been revised at p. 9 lines 208-9 to now make this point clear. |
| 8 | at line 204 and 206, why is 300km a maximum burn distance instead of 500km? | Exploratory GAMM analyses revealed that on average, beyond a distance of ca. 300 km, the influence of prescribed burns on PM2.5 concentrations at the target locations was negligible. |
| 9 | at line 220, figure 3 show increasing trends in all season after 2011. Author also mentioned that the new instruments had been used to measure PM2.5 concentration since 2012. Does the instrument change contribute the increasing trend? | Changes in measurement instrumentation have a potential for introducing systematic biases in these annual PM2.5 trends. Recently, based on the high correlation between beta attenuation monitors (BAMs) PM2.5 measurements and long-term nephelometer visibility measurements at each monitoring site, the NSW Government (2016, 2017a, 2017b) reconstructed a more consistent annual average PM2.5 time series. Their results also showed a tendency of increasing annual PM2.5 levels near 2011/2012 in some Sydney subregions, as is consistent with the results from this study. Moreover, our study also indicates that the trends start increasing from 2011 during spring and winter, which pre-dates the instrumentation change. These results suggest that the instrumentation changes that occurred in 2012 are likely to have |

| | | minimal impact the trend analysis reported in this analysis.

We have expanded the first paragraph of the Discussion to raise these points and three studies conducted by the NSW government investigating this issue are cited (please see p.13, lines 308-318 in the main text; also p.7 in Air Quality in NSW by NSW Government , 2017a). |
|---|---|---|
| 10 | at line 223, figure 4a show PM2.5 is lower in Spring than other seasons, but figure 3 does now show PM2.5 lower in Spring. Any explanation? | This comment relates to Figures 4 and 5a in the revised manuscript.

Figure 5a provides a different temporal perspective on PM2.5 concentrations than the seasonal variation shown in Figure 4. Taking the austral Autumn as an example, PM2.5 concentrations are comparatively very low in the first two months of Autumn (March and April), but then increase sharply in May. If we consider Spring, PM2.5 concentrations are higher at the start of Spring (September and October), before decreasing sharply from November as the weather dries and warms (often fewer or no HRBs occur from December to February in Sydney surroundings). |
| 11 | at line 224, what reason is the PM2.5 higher in weekend while other pollutants get lower? | There are a couple of explanations for why PM2.5 tends to be higher at weekends. First, hazard reduction burns are often conducted at weekends. Also, there may be increased domestic wood-fired heating at weekends during the colder months. These explanations are stated on p. 13 lines 319-324.

In contrast, $NO_2$ and $NO_x$ are associated with motor vehicle emissions. On the assumption that there is greater use of vehicles during the working week (e.g. commuting to and from employment) relative to during weekends, it could be expected that concentrations of these pollutants will be lower at weekends. |